# GRAPH TREE NEURAL NETWORKS

## ABSTRACT

In the field of deep learning, various architectures have been developed. However, most studies are limited to specific tasks or datasets due to their fixed layer structure. In this paper, we do not express the structure delivering information as a network model but as a data structure called a graph tree. And we propose two association models of graph tree neural networks(GTNNs) designed to solve the problems of existing networks by analyzing the structure of human neural networks. Defining the starting and ending points in a single graph is difficult, and a tree cannot express the relationship among sibling nodes. On the contrary, a graph tree(GT) can express leaf and root nodes as its starting and ending points and the relationship among sibling nodes. Instead of using fixed sequence layers, we create a GT for each data and train GTNN according to the tree's structure. GTNNs are data-driven learning in which the number of convolutions varies according to the depth of the tree. Moreover, these models can simultaneously learn various types of datasets through the recursive learning method. Depth-first convolution (DFC) encodes the interaction result from leaf nodes to the root node in a bottom-up approach, and depth-first deconvolution (DFD) decodes the interaction result from the root node to the leaf nodes in a top-down approach. To demonstrate the performance of these networks, we conducted three experiments. In the first experiment, we verified whether it can be processed by combining graph tree neural networks and feature extraction networks(combining association models and feature extraction models). The second experiment is about whether the output of GTNN can embed information on all data contained in the GT(association). In the third experiment, we simultaneously learn the structure of images, sounds, graphs, and trees structure datasets (processing various datasets). We compared the performance of existing networks that separately learned image, sound, and natural language, relation datasets with the performance simultaneously learned by connecting these networks. As a result, these models learned without significant performance degradation, and the output vector contained all the information in the GT. And the importance of transfer-learning and fine-tuning in these models is explained.

## 1 INTRODUCTION

In deep learning, various architectures have been designed such as CNN, RNN, and GNN(LeCun et al., 1989; Hopfield, 1982; Scarselli et al., 2008; Wu et al., 2020). These networks show good performance in various fields such as image, text, and sound etc. However, most of the existing studies are focused on a specific dataset or task. And there are many tasks that humans can do but are difficult to perform with neural networks.

First, some of the difficulties with imitating human neural networks are introduced below. (C-i) Sensory organs that process information for each data type exist at several starting points. (C-ii) A tree cannot express the relationship between sibling nodes, and it is difficult to define the graph's starting point and ending point. (C-iii) The structure that receives information is hierarchical and freely integrated with various structures. But existing networks are designed to learn only specific tasks or datasets because layers are fixed. (C-iv) The number of activated neurons and the processing depths differ depending on the data type and complexity. And sometimes the information is entered and sometimes not. (C-v) Human neurons are very numerous and process various types of data.

C-[·] denotes the characteristics. (C-i) Humans have various sensory organs from which they receive information such as sight, hearing, and smell, and transfer it to the cerebral cortex using different

information-processing organs. The optic nerve also begins with the visual cortex of the occipital

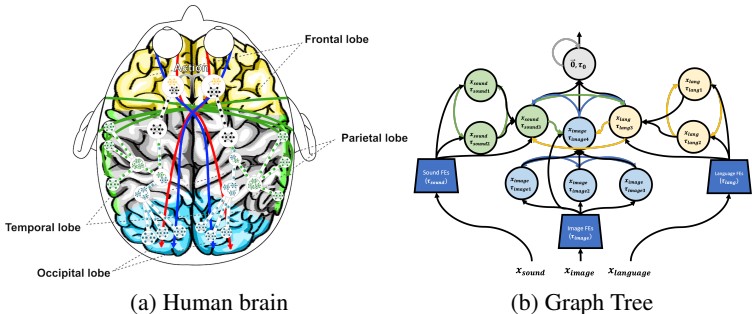

(a) Human brain        (b) Graph Tree

Figure 1: Comparison of the human brain and graph tree neural network

lobe, and the auditory nerve begins with the auditory cortex of the temporal lobe; this means that the two paths have different starting points. (C-ii) A graph can describe the relationship depending on whether all nodes are connected, but it is not easy to define the start and end points. A tree is a data structure that can define direction with the leaf and root nodes. On the other hand, a tree structure cannot express the relationship between sibling nodes. (C-iii) Information from different sensory organs is gathered in the association area. Typically, the posterior parietal cortex(Malach et al., 1995) is located at the top of the head as part of the parietal lobe, and sensory information such as vision and hearing is fused and interpreted; Then, information is coordinated in the frontal lobe and humans act. (C-iv) There is also an over-fitting problem in DNN(Widrow et al., 1960), and various attempts have been made to solve this in (Srivastava et al., 2014). In DNN, simply deeper layers lead to an over-fitting problem(Dai et al., 2017). In particular, NasNet(Zoph et al., 2018) has attempted to design a layer automatically through reinforcement learning and RNN. This means that there is an appropriate network depth depending on the complexity of the dataset; we want to design a network that could adjust the depth of the layers according to the complexity of each data, not the data set. (C-v) Also, the number of human neurons is about 85 billion(Herculano-Houzel, 2009), which is difficult to express in a network. Therefore, GTNNs are designed to solve these problems.

A-[·] denotes our approach of C-[·]. (A-i) This network can perform information processing, such as that done by sensory organs before their signals are entered into the cerebral cortex. In this paper, we express this sensory organ as a feature-extraction process(sec.2.2). Type information is stored with the data and converted into an input vector by different feature extraction processes for each type. (A-ii) We propose a new data structure called Graph Tree Node(GTN) and Graph Tree(GT). A graph tree can be expressed using leaf and root nodes as the starting and ending points, as well as the relationship among sibling nodes. In addition, we can modify the existing tree dataset at no high cost. (A-iii) subtrees of different types are merged and make a final decision at the root node; this structure is difficult to express with the existing forward-learning method. Therefore, we used a recursive-learning method(Goller & Kuchler, 1996). This methodology has been effective in analyzing the semantics of programming source code or natural language(Socher et al., 2013; Mou et al., 2014). We call the recursive-convolution methodology used in GTNNs the depth-first convolution(DFC) methodology(sec.2.3). DFC is a convolution method in which subtrees originating from different Leaf nodes are integrated into a bottom-up approach and represent hierarchical and relational information. GTNNs can simultaneously learn various types of datasets through the recursive learning method because the graph tree data structure can express various model structures. (A-iv) GTNNs are data-driven learning in which the number of convolutions varies according to the depth of the tree. Instead of using fixed sequence layers, we create a graph tree for each data and learn according to the tree's structure. If we modify this network further, we can adjust the depth according to the type and complexity of each data. In addition, We introduce two models of GTNNs called graph tree convolutional networks(GTC) and graph tree recursive networks(GTR), and these models are mathematically related to MLP(Hinton et al., 2006) and RNN. The Fully Connected layer(FC layer) in MLP as Level Layer of GTC, and Time in RNN as depth of GTR on a special case. (A-v) The number of human neurons is enormous, and it is difficult to express all of their structures in a network; on the other hand, this network can represent multiple neurons because it can perform recursive end-to-end learning according to the amount of information.

This network theory holds that "units of information have a relationship in the form of a graph, then become a bigger unit of information, and have a relationship with other units of information. At this point, the unit of information is a set of neurons, and we can express it as a vector with GTNN."

To demonstrate the performance of this network, we conducted three experiments. And we used several benchmark datasets (image(MNIST(LeCun et al., 1998)), sound(Speech Commands(Warden, 2018)), text(IMDB(Maas et al., 2011),SST(Socher et al., 2013)), graph(Dou et al., 2021)).

In the first experiment, It verifies whether feature extraction networks and association networks can be learned together. we compared the performance of existing networks that separately learned image(LeNet-5(LeCun et al., 1989), sound(M5(Dai et al., 2017)), text(CNN(Kim, 2014)) with the performance simultaneously learned by connecting these networks for feature extraction into GTNN.

The second experiment contained one or three types of data into a GT, and GTNNs learned these GTs. Then, we verified whether the output contained all the information inside the GT. Then, we conducted a verification experiment on whether the network's output contained all information on the tree.

In the third experiment, we verifies whether data of various structures(image, sound, tree, graph) can be learned and checked the plot of the learning process in appendix E. As a result, the network was learned without significant degradation in performance compared to when learning existing networks separately, and all information on the tree could be embedded as a vector.

## 2 GRAPH TREE NEURAL NETWORKS

In this section, we proposes graph tree architecture. The architecture consists of three parts

(i) Defining the data structure of graph tree node(GTN) and designing the graph tree structure (GT). (ii) Defining the feature-extraction(FE) process. (iii) Defining the GTNN model.

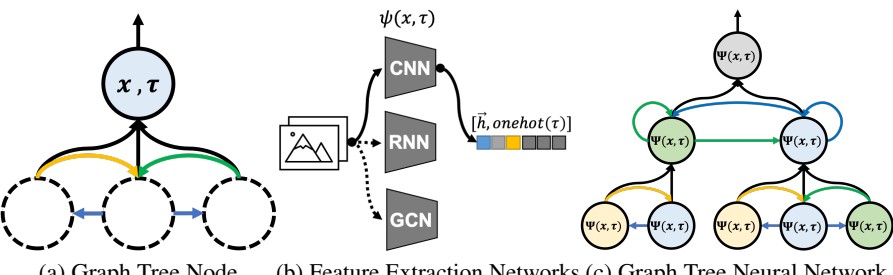

(a) Graph Tree Node     (b) Feature Extraction Networks     (c) Graph Tree Neural Network

Figure 2: Graph Tree Architecture

### 2.1 GRAPH TREE DATA STRUCTURE

The graph tree (GT) is a data structure that can express relational and hierarchical information.

- $x$ : (Input) – we can store the data such as images, sound, text, or tabular to each node.
- $\tau$ : (Type) – This refers to the type of information matching the input $x$. the feature-extraction function($\psi$) is selected by $\tau$ like Eqn.1.
- $\mathbf{A}_c$ : (Children Adjacency Matrix) – This refers to the relationship information that exists in GTN. the number of children is $N$ and we can express it as $\mathbf{A}_c \in \mathbb{R}^{N \times N}$.
- $\mathbf{C}$ : (Children) – This refers to the child nodes of GTN matching to the node of $\mathbf{A}_c$. If there is no matching child, it is replaced with $\mathbf{GTN}_\emptyset$, which carries the initial hidden state $\overrightarrow{0}$ instead. We can express this as $\mathbf{C} = \{\mathbf{GTN}_1, \mathbf{GTN}_2, ..\mathbf{GTN}_N\}$.

We can express $\mathbf{GTN} = \{x, \tau, \mathbf{A}_c, \mathbf{C}\}$, $\mathbf{GTN}_i \in \mathbf{GT}$. $\mathbf{GTN}_{root}$ denotes the root node of $\mathbf{GT}$. The reason for defining the relationship among child nodes is the convenience of implementation. If we define GTN in the above way, we can convert the tree dataset into the GT dataset without significantly modifying the tree structure that has previously been useful.

## 2.2 Feature Extraction networks & Type Bias

$$\overrightarrow{x} = \Psi(x, \tau) = [\psi_\tau(x), onehot(\tau)] \tag{1}$$

The $x$(input) and $\tau$(type) exist together in GTN, and the feature extraction process of $x$ is selected by $\tau$ as Eqn 1. We expressed the function of feature extraction as $\Psi$, and this function converts $x$ to $\overrightarrow{x}$. And the one-hot vector has the effect of having a different bias value for each type($\tau$). The weight parameter corresponding to the type one-hot vector means the bias value for the corresponding type, and the activation threshold value is adjusted for each type-bias. It can represent data type information and that it is transmitted from which nerve. In addition, we do not need to perform adding operations(+) for bias. Therefore, the empty space of GT can be used as a zero-vector when batch learning is performed. And the dictionary structure is very useful for batch learning; we attached the methodology to appendix.A. This methodology divides mini-batch samples by type, and the type-mini-batch size varies each time. Therefore, we recommend using a weight standardization(Qiao et al., 2019) or group normalization(Wu & He, 2018) to avoid batch normalization(Ioffe & Szegedy, 2015) affected by the batch size. We described the details of the network we used in appendix.C.

## 2.3 Depth-first Convolution & Deconvolution

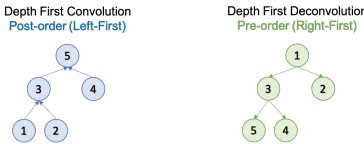

Figure 3: Depth First Convolution & Deconvolution

This section introduces two propagation methods of depth-first convolution (DFC) (a convolution method for traversing all nodes of GT) and depth-first deconvolution (DFD). Depth-first search (DFS) is a search algorithm for the tree data structure. Unlike general recursive convolution, DFC and DFD are methodologies for learning children's relationships, feature extraction networks together.

---

**Algorithm 1** Depth First Convolution

---

1: **function** DFC($\mathbf{GTN}_p, lv$)
2: $\quad x_p, \tau_p, \mathbf{A}_{cp}, \mathbf{C}_p \leftarrow \mathbf{GTN}_p.items()$
3: $\quad \overrightarrow{x}_p, *i_1 \leftarrow \Psi[\tau_p](x_p)$
4: $\quad ChildrenList \leftarrow [\,]$
5: $\quad$ **for** $q \leftarrow 1...N_p$ **do** $\qquad \triangleright$ left-first
6: $\qquad \overrightarrow{x'}_{pq}, \overrightarrow{h'}_{pq} \leftarrow$ DFC($\mathbf{C}_p[q], lv + 1$)
7: $\qquad ChildrenList.append([\overrightarrow{x'}_{pq}, \overrightarrow{h'}_{pq}])$
8: $\quad$ **end for**
9:
10: $\quad$ **if** len($\mathbf{C}_p$) is 0 **then** $\qquad \triangleright$ leaf node
11: $\qquad \mathbf{GTN}_p.more \leftarrow (*i_1)$
12: $\qquad$ **return** $[\overrightarrow{x}_p, \overrightarrow{0}]$
13: $\quad$ **end if**
14: $\quad [\mathbf{x'}_p, \mathbf{h'}_p] \leftarrow$ Stack($ChildrenList$)
15: $\quad \overrightarrow{h}_p, i_2 \leftarrow \mathbf{GTNN}(\mathbf{A}_c, \mathbf{x'}_p, \mathbf{h'}_p, lv)$
16: $\quad \mathbf{GTN}_p.more \leftarrow (*i_1, *i_2)$
17: $\quad$ **return** $[\overrightarrow{x}_p, \overrightarrow{h}_p]$
18: **end function**

---

**Post-order Depth-first convolution (Left-First)** If DFS is post-order, iteration starts from the leaf node. Likewise, DFC is a recursive convolution that propagates from the deepest nodes. GTNN Utilizes the hidden vector expressing the current depth and state.

First, we need to bring four items($x_p, \tau_p, A_{cp}, C_p$) of the current $\mathbf{GTN}_p$. the p means the convolution order, and $x_p$ is embedded by the feature extraction network($\Psi$) according to the type($\tau_p$) and becomes the feature vector($\overrightarrow{x}_p$).

Graph tree network models perform the convolution by receiving information($\overrightarrow{x'}_{pq}, \overrightarrow{h'}_{pq}$) from the current node's children and the relationship($A_{cp}$) among children's information. We can use this process like learning the relationship between siblings nodes. the $\overrightarrow{h}_p$ means the convolution output of the child nodes. finally, $\overrightarrow{x}_p$ and $\overrightarrow{h}_p$ are delivered to the parent node. If the child node does not exist($\mathbf{C}_p$), the current node means a leaf node. Therefore, the feature vector($\overrightarrow{x}_p$) and initial hidden state($\overrightarrow{0}$) are transmitted to the parent node.

The important thing is that the root node only receives information from their children and does not the convolution with the root feature vector. Therefore, like algo.3, convolution is performed once more on the output of the DFC. If we repeat this process, we finally get $\overrightarrow{h}_{root}$. Additionally, we can store information($*i_1, *i_2$) generated during the DFC process in $\mathbf{GTN}.more$ and use it during the DFD process (ex. If the aggregate function is a maxpool, we can use the indices information during deconvolution).

---

**Algorithm 2** Depth First Deconvolution

---

1: **function** DFD($\tilde{x}_p, \tilde{h}_p, \mathbf{GTN}_p, *lv$)
2:     $\tau_p, \mathbf{A}_{cp}, \mathbf{C}_p \leftarrow \mathbf{GTN}_p.dconv\_items()$
3:     $*i_1, *i_2 \leftarrow \mathbf{GTN}_p.more$
4:     $\mathbf{GTN}_p.\hat{x}_p \leftarrow \Psi^{-1}[\tau_p](\tilde{x}_p, *i_1)$
5:     **if** len($\mathbf{C}_p$) is 0 **then**     ▷ leaf node
6:         **return**
7:     **end if**
8:     $\hat{\mathbf{A}}_{cp}, \tilde{\mathbf{x}}'_p, \tilde{\mathbf{h}}'_p \leftarrow \mathbf{GTNN}^{-1}(*\mathbf{A}_{cp}, \tilde{h}_p, lv, *i_2)$
9:     $\mathbf{GTN}_p.\hat{\mathbf{A}}_c \leftarrow \hat{\mathbf{A}}_{cp}$
10:     **for** $q \leftarrow N_p...1$ **do**     ▷ right-first
11:         $\tilde{x}'_{pq}, \tilde{h}'_{pq} \leftarrow \tilde{\mathbf{x}}'_p[q], \tilde{\mathbf{h}}'_p[q]$
12:         DFD($\tilde{x}'_{pq}, \tilde{h}'_{pq}, \mathbf{GTN}_p.\mathbf{C}[q], lv + 1$)
13:     **end for**
14:     **return**
15: **end function**

---

**Pre-order Depth-first Deconvolution (Right-First)** DFD is a decoding methodology for GTNNs. The DFC propagates from the leaf node to the root node, whereas the DFD propagates from the root node to the leaf node in the order of pre-order depth first to decode this.

First, we need to bring three items($\tau_p, A_{cp}, C_p$) of the current $\mathbf{GTN}_p$. $\tilde{x}_p$ is decoded by the feature decoder network($\Psi^{-1}$) according to the type($\tau_p$) and the decoded output is stored in the $\mathbf{GTN}_p.\hat{x}_p$.

If the child does not exist, this means a leaf node and performs a return. Otherwise, The hidden state($\tilde{h}_p$) is restored to $\tilde{\mathbf{x}}'_p, \tilde{\mathbf{h}}'_p$ through $\mathbf{GTNN}^{-1}$; and $\tilde{\mathbf{h}}'_p$ is transferred to child nodes. Also, if DFC proceeds from left to right as left first ($1...N$), DFD proceeds from right to left as right first ($N...1$). Please see Fig.3.

---

**Algorithm 3** Propagate (for AutoEncoder Models)

---

1: **function** PROPAGATE($\mathbf{GT}_{root}$)
2:     $[\overrightarrow{x}'_{root}, \overrightarrow{h}'_{root}] \leftarrow$ DFC($\mathbf{GTN}_{root}, 1$)
3:     $\overrightarrow{h}_{root}, *i_2 \leftarrow \mathbf{GTNN}(\mathbf{I}, \overrightarrow{x}'_{root}, \overrightarrow{h}'_{root}, 0)$
4:     $*\_, \tilde{x}'_{root}, \tilde{h}'_{root} \leftarrow \mathbf{GTNN}^{-1}(\mathbf{I}, \overrightarrow{h}_{root}, 0, *i_2)$
5:     DFD($\tilde{x}'_{root}, \tilde{h}'_{root}, \mathbf{GTN}_{root}, 1$)
6:     **return** $\overrightarrow{h}_{root}, \mathbf{GTN}_{root}$
7: **end function**

---

Like the propagate function (algo.3), we can freely encode and decode through DFC and DFD. We want to show the relationship between DFC and DFD and the possibility of expanding to the widely used autoencoder(Hinton & Salakhutdinov, 2006) models. In addition, Implementing this recursive function as a loop(while) will speed up.

## 2.4 ASSOCIATION NEURAL NETWORK MODELS : GTR & GTC SERIES

This section proposes GTR and GTC series that perform inductive learning in GTNNs. We describe the convolution performed in one graph tree node by DFC. These networks can be used as an association cell or layer, and we explained the difference between the two models in appendix B.

These association models do not use any fixed architecture and integrate information according to the GT structure. And by fusing graph and tree structures, we can learn more diverse data structures with one cell and various architectures can be expressed by data(please see the appendix.D). Accordingly, these models are freer to perform various tasks and learn a lot of information using fewer parameters.

These association models are as follows: The feature extraction models create the information from data; It is the "smallest unit" of information in the GT. The parent GTN receives information from their children and perform the convolution operation with the relation term; the convolution output is a "bigger unit" of information. Finally, a root GTN has all GT information and perform the convolution operation; the output is "the biggest unit" of information.

### 2.4.1 ASSOCIATION CELL : GRAPH TREE RECURSIVE NEURAL NETWORK

We will describe a GTR cell that trains the GT described previously in Sec 2.1. GTR model can be expressed as $\mathbf{GTR} = \{\mathbf{W}, \Psi, g, \sigma\}$ $\mathbf{W} \in \mathbb{R}^{F' \times (F+B+F')}$. $F$ is the node-feature size, $B$ is the type-bias size, and $F'$ is the output and hidden size, which is the information received from the child node. $\Psi$ denotes the feature-extraction(Eqn 1) process described above, $g$ denotes the aggregate function, and $\sigma$ denotes the activation function.

$$\overrightarrow{x}_p = [\psi_{\tau_p}(x_p), onehot(\tau_p)] = \Psi(x_p, \tau_p), \overrightarrow{x}_p \in \mathbb{R}^{F+B} \tag{2}$$

$$\overrightarrow{h}_p = g(\sigma(\tilde{\mathbf{D}}_{cp}^{-\frac{1}{2}}\tilde{\mathbf{A}}_{cp}\tilde{\mathbf{D}}_{cp}^{-\frac{1}{2}}(\mathbf{W}(\|_{q=0}^{N_p}[\overrightarrow{x'}_{pq}, \overrightarrow{h'}_{pq}]^T)^T))) \tag{3}$$

The propagation for GTNN is proceeding by the DFC traversing from leaf nodes to a root node (sec.2.3). $p$ is the order of DFC, $q$ is the child node number, $N_p$ is the number of children of $\mathbf{GTN}_p$, $x_p$ is the node-input of $\mathbf{GTN}_p$.

The $\Psi$ extracts the feature vector($\overrightarrow{x}_p$) from $x_p$ of $\mathbf{GTN}_p$ through the feature extraction process(Eqn.2) considering the type information($\tau_p$). The first location to start is leaf nodes by DFC. In the Leaf node, the initial hidden state is a zero vector($\overrightarrow{0}$), and this is concatenated with $\overrightarrow{x}_p$ to become $[\overrightarrow{x}_p, \overrightarrow{h}_p] \in \mathbb{R}^{F+B+F'}$ through the concatenation process as $[,]$ and then delivered to the parent node. If a child node exists, the current node receives information from a child node as $[\overrightarrow{x'}_{pq}, \overrightarrow{h'}_{pq}]$. $\|_{q=0}^{N_p}[\overrightarrow{x'}_{pq}, \overrightarrow{h'}_{pq}]^T$ means that $[\overrightarrow{x'}_{pq}, \overrightarrow{h'}_{pq}]$ matching the graph nodes($\mathbf{A}_{cp}$) existing in the p-th node are stacked as much as $N_p$, which is $\|_{q=0}^{N_p}[\overrightarrow{x'}_{pq}, \overrightarrow{h'}_{pq}]^T \in \mathbb{R}^{N_p \times (F+B+F')}$.

We applied the GCN methodology(Kipf & Welling, 2016), which is useful in the GNN field. Therefore we expressed as $\tilde{\mathbf{A}}_{cp} = \mathbf{A}_{cp} + \mathbf{I}$, where $\mathbf{I}$ is the Identity matrix. If we express the connection value as 1, just the more connected nodes becames the scale value larger than others. Therefore, $\tilde{\mathbf{D}}_{cp}^{-\frac{1}{2}}\tilde{\mathbf{A}}_{cp}\tilde{\mathbf{D}}_{cp}^{-\frac{1}{2}}$ is applied using the order matrix $\tilde{\mathbf{D}}_{cp}$ of $\tilde{\mathbf{A}}_{cp}$ as a method of normalizing the relationship matrix in Eqn. 3. we used $F$ as 128, $B$ as 3(image, sound, language), and $F'$ as 128, and $\sigma$ as ReLU(Nair & Hinton, 2010), and g was the readout of Max. If there is no sibling node of the current node, GTR is mathematically identical to the RNN as $\mathbf{IW}[\overrightarrow{x}, \overrightarrow{h}]$. RNN can be a special case of GTR, and the depth of the tree means times of the RNN.

### 2.4.2 ASSOCIATION LAYERS : GRAPH TREE CONVOLUTIONAL NEURAL NETWORK

GTC is composed of $\mathbf{GTC} = \{\{\mathbf{W}_0, ...\mathbf{W}_m\}, \Psi, g, \sigma\}$. There is $\mathbf{W}_{lv} \in \mathbb{R}^{F'_{lv} \times (F_{lv}+B+F'_{lv+1})}$ in charge of each level, and the network in charge of each level is called the level layer. The input size of the level is $F_{lv}$, its output size is $F'_{lv}$, and the output size of the child is $F'_{lv+1}$. Therefore, it is possible to adjust the input, hidden size; and it can learn the depth-limited tree, $m$ means Maximum-Depth. We set $F_{lv}$ and $F'_{lv}$ to 128 for all lvs in the same way as GTR in the experiment. If the input size of $F_{lv}$ is 0 and there is no sibling node of the current node, this network is mathematically identical to the FC layer as $\mathbf{IW}_{lv}\overrightarrow{x}^T$ and the MLP as $(\mathbf{IW}_0..(\mathbf{IW}_{lv-1}(\mathbf{IW}_{lv}\overrightarrow{x}^T)))$, $\mathbf{I}$ is the Identity matrix. The mathematical expression of GTC is similar to Eqn 3. The depth of the tree indicates the number of layers of the MLP. Therefore, the MLP and FC layer can be a special case of GTC.

## 2.5 ATTENTION MODELS

There is an attention process in the human brain. Therefore, GATs(Veličković et al., 2017), a model that has been useful recently, was combined.

### 2.5.1 GRAPH TREE RECURSIVE ATTENTION NETWORKS

We introduce GTRAs that learn the importance through attention by slightly modifying the expression of the GTR. It is composed of $\mathbf{GTRAs} = \{\mathbf{W}, \Psi, g, \sigma, \sigma_a, \overrightarrow{a}\}$ that added $\{\sigma_a, \overrightarrow{a}\}$ in GTR. A parameter for attention mechanism is added ($\mathbb{R}^{2F'} \times \mathbb{R}^{2F'} \to \mathbb{R}$) and the attention's activation function used LeakyReLU(Xu et al., 2015) in the same way as GATs. $\mathcal{N}_{pq}$ is a set of nodes connected

to the q-th child's node in the $A_{cp}$ of $\mathbf{GTN}_p$, and we can express this as:

$$\alpha_{pqr} = \frac{\exp(LeakyReLU(\overrightarrow{\mathbf{a}}^T[\mathbf{W}[\overrightarrow{x'}_{pq}, \overrightarrow{h'}_{pq}]^T, \mathbf{W}[\overrightarrow{x'}_{pr}, \overrightarrow{h'}_{pr}]^T]))}{\sum_{k \in \mathcal{N}_{pq}} \exp(LeakyReLU(\overrightarrow{\mathbf{a}}^T[\mathbf{W}[\overrightarrow{x'}_{pq}, \overrightarrow{h'}_{pq}]^T, \mathbf{W}[\overrightarrow{x'}_{pk}, \overrightarrow{h'}_{pk}]^T]))} \tag{4}$$

With the attention methodology introduced in GATs(Veličković et al., 2017), it learns how the r-th node is of importance to the q-th node. This information is replaced with the part to which the adjacency matrix is connected.

If $\mathbf{A}_{cp}$ does not exist in the GTN, we utilized to select critical features as :

$$score_{pqr} = \begin{cases} Sigmoid(\overrightarrow{\mathbf{a}}^T[\mathbf{W}[\overrightarrow{x'}_{pq}, \overrightarrow{h'}_{pq}]^T, \mathbf{W}[\overrightarrow{x'}_{pr}, \overrightarrow{h'}_{pr}]^T]) & \text{if } q \neq r \\ 1 & \text{if } q = r \end{cases} \tag{5}$$

$$\alpha_{pqr} = \frac{\exp(score_{pqr})}{\sum_{k=1}^{N_{pq}} \exp(score_{pqk})} \tag{6}$$

The self-node is set to 1 and the relationship values of the node are replaced with a real value of $0 \sim 1$ using Sigmoid. Therefore, we can train the attention learning without $\mathbf{A}_{cp}$. It means to learn relationships based on self-node. Other useful models is self-adaptive model(Wu et al., 2019).

Therefore, the GTRAs are as follows:

$$\overrightarrow{h}_p = g(\sigma((\mathbf{A}_{cp} \odot \alpha_p)(\mathbf{W}[\mathbf{x}'_p, \mathbf{h}'_p]^T))) \tag{7}$$

In Eqn.7, $\odot$ indicates point-wise and $\mathbf{x}'_p$ is a stack of $\overrightarrow{x'}_{pq}$ as $\|_{q=0}^{N_p} \overrightarrow{x'}_{pq}^T$ and it is $\mathbf{x}'_p \in \mathbb{R}^{N_p \times (F+B)}$. $\mathbf{h}'_p$ is a stack of $\overrightarrow{h'}_{pq}$ as $\|_{q=0}^{N_p} \overrightarrow{h'}_{pq}^T$ and $\mathbf{h}'_p \in \mathbb{R}^{N_p \times F'}$. Therefore it is $[\mathbf{x}'_p, \mathbf{h}'_p] \in \mathbb{R}^{N_p \times (F+B+F')}$.

To further stabilize the self-attention process, we introduce a multi-head attention mechanism:

$$\overrightarrow{h}_p = g(\|_{k=1}^K \sigma(\mathbf{A}_{cp} \odot \alpha_p^k)(\mathbf{W}^k[\mathbf{x}'_p, \mathbf{h}'_p]^T)) \tag{8}$$

where $K$ is the number of multi-heads. The results from multiple heads are concatenated and delivered to a parent node. We set K to 8 and set output dim for each head to 16($=F'_{lv}/K$). therefore we set $F'_{lv}$ to 128. This process becomes a cell and delivers the result from the leaf node to the root node.

### 2.5.2 GRAPH TREE ATTENTION NETWORKS

The GTAs model is modified from GTC, and GATs' attention mechanism was applied. this model can be expressed as $\mathbf{GTAs} = \{\{\mathbf{W}_0, ...\mathbf{W}_m\}, \{\overrightarrow{a}_0, ...\overrightarrow{a}_m\}, \Psi, g, \sigma, \sigma_a\}$, and $\{\sigma_a, \overrightarrow{a}_{lv}\}$ are added in GTC($\overrightarrow{a_{lv}} \in \mathbb{R}^{2F'_{lv}}$). Unlike GTRAs, GTAs can design different sizes of $F, F'$ to match the lv. this model is designed to select critical features, and it is similar to Sec.2.5.1

**Supervised learning**  Supervised learning uses datasets with labels. In this network, the $\overrightarrow{h}_{root}$ was mapped to the dimension of the class using a fully connected layer($F'$ to class count) and we calculated the log softmax and negative log likelihood loss for supervised learning as:

$$\hat{y} = log\_softmax(TaskLayer(\overrightarrow{h}_{root})) \tag{9}$$

$$loss = negative\_log\_likelihood\_loss(y, \hat{y}) \tag{10}$$

## 3 EXPERIMENTAL RESULTS

This section introduces the contents of the experiment and the GT dataset consisting of the image, sound, and natural language. How to design GT is introduced in appendix.D.

**Datasets**  Our goal is to express the information delivery process of existing models as a graph tree of data. The first experiment is whether various datasets can be simultaneously learned by combining GTNN and feature extraction networks. the GT we used in the experiment is illustrated by Fig.4(a). Layer node means that it has no input value, only performs convolution. If there is no sibling node, GTC performs the same operation as one FC layer. Sound and natural language GT have two nodes, and image GT has three nodes because It was configured similarly to figure (likes this fig.1). The

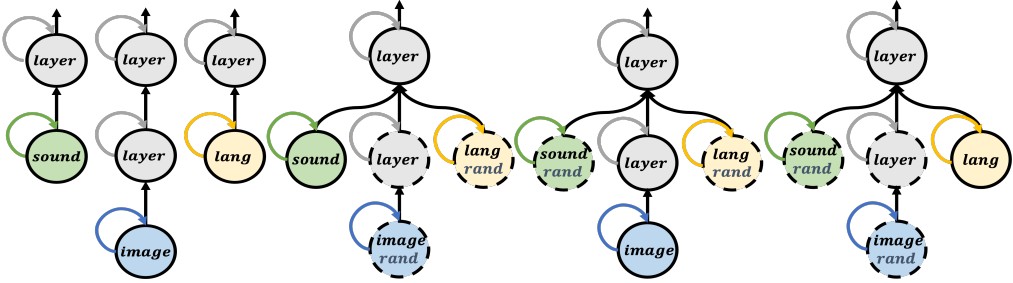

(a) Expt 1,2: One type   (b) Expt 2: All to sound   (c) Expt 2: All to image   (d) Expt 2: All to language

Figure 4: Graph tree datasets of the experiments

important point is not limited to any specific architecture; later, this GT could be modified according to the task or complexity.

The second experiment is about whether the output($\overrightarrow{h}_{root}$) of GTNN is embedded information on all data contained in the GT(association). Therefore, we performed image, sound, and text classification by putting image, sound, text data in one GT, and we also learned the GT used in the first experiment simultaneously. One problem is that when the three types of data are combined, The number of combinations of the dataset becomes too huge. For example, there are 50,000 images, 105,000 sounds, and 17,500 natural language samples, and when these data are combined, we need to generate 50,000×105,000×17,500 GT samples. For this problem, we used a sampling method. When loading each data, two data of different types are sampled randomly. Thus, when we load one data, two GTs are generated of the one type GT and the three type GT. The test accuracy is the result of averaging 5-fold. the GT dataset of the second experiment is illustrated by Fig.4(a,b,c,d). In the third experiment, we added the graph and tree structure dataset, This is described in Appendix E.

## 3.1   THE EXPERIMENT 1 : INDIVIDUAL & SIMULTANEOUS (PROCESSING VARIOUS DATASETS)

Table 1: the result of Experiment 1

| Model | Task layers | | | 47(=10+35+2) class | | | Transfer learning | | | Fine tuning | | |
|---|---|---|---|---|---|---|---|---|---|---|---|---|
| | MNIST | SC | IMDB | MNIST | SC | IMDB | MNIST | SC | IMDB | MNIST | SC | IMDB |
| LeNet-5 | 98.46 | - | - | 98.46 | - | - | 98.49 | - | - | 98.49 | - | - |
| M5 (Group Norm) | - | 97.63 | - | - | 97.63 | - | - | 100.0 | - | - | 100.0 | - |
| CNN | - | - | 87.13 | - | - | 87.13 | - | - | 87.48 | - | - | 87.48 |
| GTC | 98.84 | 97.14 | 86.10 | 98.58 | 96.87 | 85.88 | 98.39 | 98.81 | 82.27 | 98.87 | 99.07 | 86.99 |
| GTAs | **98.96** | 96.91 | 85.38 | 98.77 | 96.83 | 85.76 | **98.65** | 98.91 | 82.20 | 98.64 | 97.39 | 86.62 |
| GTR | 98.53 | 96.54 | 85.62 | **98.79** | 97.07 | 86.24 | 98.48 | 99.66 | 81.91 | **98.94** | 99.17 | 87.18 |
| GTRAs | 98.79 | 96.66 | 86.28 | 98.78 | 96.68 | 85.79 | 98.47 | 98.64 | 82.18 | 98.92 | 98.89 | 87.27 |
| FE Epochs | 30 | 30 | 30 | 30 | 30 | 30 | 26 | 95 | 3 | 26 | 95 | 3 |

We trained without using a pre-trained model to check whether various feature extraction models can be learned simultaneously. Only word embedding network(Pennington et al., 2014) was used as a pretrained model. Existing networks learn only about specific datasets. Therefore, LeNet-5(LeCun et al., 1998), M5(Dai et al., 2017), and CNN(Kim, 2014) were selected for feature extraction networks of the image(MNIST(LeCun et al., 1998)), sound(Speech Command(Warden, 2018)), and natural language(IMDB(Maas et al., 2011)). We described more details in appendix C. The reason for choosing these networks is based on Convolutional Neural Networks specialized extract features.

Since the number of classes is different for each data, the test was divided into two cases. (Task Layers): The last layer was placed differently for each dataset and mapped to the class dimension of the dataset. (47 class): The one last layer was used for all datasets, and the number of class dimensions is equal to the sum of class dimensions of all datasets. Because if the last layer is placed differently for each dataset, information on the input type may be informed. All models were learned during 30 epochs, and the results were as Table.1(Task layers, 47 class), and the learning process is (Fig.5(a)).

The learning processes of GTNNs are similar to the process of individually learning feature extraction networks; if the model learns like this, several problems are found. LeNet-5 is well trained, but M5

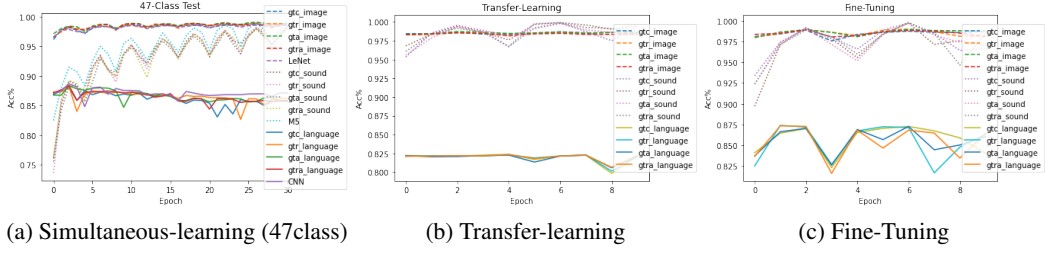

| (a) Simultaneous-learning (47class) | (b) Transfer-learning | (c) Fine-Tuning |

Figure 5: Test Accuracy (The Experiment 1)

needs to be more trained, and CNN has an overfitting problem. The reason is that each network has different epochs to have optimal performance, and setting learning parameters is more complex than learning individually.

**transfer learning & fine-tuning**    Therefore, after learning individually, we combined the association model to perform transfer-learning and fine-tuning(Pan & Yang, 2009). The result is Table.1(Transfer-learning, Fine-tuning). In transfer learning, the parameters of feature extraction networks are not modified, and in fine-tuning, the parameters are modified. We used the validation set to learn each feature extraction network and adopt the lowest value of the validation loss to use the model. As a result, the LeNet-5, M5, CNN were pre-trained with 26, 95, 3 epochs. These models were combined with GTNN to perform fine-tuning and transfer learning. Then, the network was re-learned, and learning was stopped at epochs with the lowest value of validation loss using each of the same validation datasets. As in the results of Table.1(Transfer-learning, Fine-tuning) and Fig.5(b,c), in the case of transfer learning, performance was poor in the IMDB dataset. We thought it was overfitting because the IMDB dataset was relatively small. We reduced the learning rate from 0.001 to 0.0001, but the results were similar. While fine-tuning was similar to the performance learned individually in all datasets. And the performance improved compared to when fine-tuning was not performed. Consequently, If transfer learning and fine tuning are used by setting parameters well according to the network, the performance of the network can be improved. In Experiment 3, we learn datasets of various structures(Image, sound, tree, graph) through GTNN. It is described in Appendix.E

### 3.2    THE EXPERIMENT 2 : CAN ALL DATA INFORMATION BE CONTAINED? (ASSOCIATION)

Table 2: the result of association test

| Model | Task 2 + task layer (%) | | | | | | task 2 + fine-tuning (%) | | | | | |
|---|---|---|---|---|---|---|---|---|---|---|---|---|
| | MNIST | MNIST&All | SC | SC&All | IMDB | IMDB&All | MNIST | MNIST&All | SC | SC&All | IMDB | IMDB&All |
| GTC | 98.90 | 98.88 | 96.83 | 95.71 | **87.20** | 87.18 | 98.63 | 98.62 | 98.83 | 97.66 | 86.76 | 86.61 |
| GTAs | 98.87 | 98.88 | 97.30 | 96.37 | 86.73 | **87.41** | 98.93 | **98.87** | 98.67 | 98.52 | 87.43 | 87.30 |
| GTR | **99.05** | **99.04** | 96.46 | 97.03 | 87.16 | 87.21 | **98.97** | 98.84 | 98.83 | 98.58 | 87.00 | 87.14 |
| GTRAs | 98.95 | 98.96 | 97.36 | 95.76 | 86.55 | 87.08 | 98.82 | 98.73 | 98.60 | 98.22 | 87.30 | 87.26 |
| FE Baseline | 98.46 | 98.46 | **97.63** | **97.63** | 87.13 | 87.13 | 98.49 | 98.49 | **100.0** | **100.0** | **87.48** | **87.48** |

We constructed the GT dataset described above(Fig.4(a,b,c,d)) to validate if the output vector can contain all the information in the GT, and the results are as follows Table.2. The meaning of FE Baseline is the performance of networks of LeNet-5, M5, and CNN. In these experiments, it has been verified that it is possible to learn various types of datasets using one network cell and that information can be embedded together. Consequently, we can share an association cell or layers to learn without being limited to the dataset type and embed all information inside the GT into a vector.

## 4    CONCLUSION

We introduced a data-driven network that can jointly learn relationships and hierarchical information. This study is to be free from architecture and has been developed to connect various types of information being developed. And The author of this paper believes that one day a neural network will emerge that can perform all human tasks. In addition, this is an association model that behaves like human sensory organs and can be described as similar to a human neural network. We think this network can be used as a deep neural network part of DQN(Mnih et al., 2013). We will try to leverage this network to approach problems that have not been solved before. This paper is part of a series; in the next paper, we introduce Graph Tree Deductive Networks(GTDs).

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

## A  Type Embedding Networks for Batch Training

---
**Algorithm 4** Batch Type Embedding
---
1: **function** TYPEEMBEDDING($GT_{batch}$)
2:  $dict_x = \{\}$                        $\triangleright key(\tau) : value(list_x)$
3:  $dict_{idx} = \{\}$               $\triangleright key(batch\_idx) : value(\tau, index_{dict_x})$
4:  **for** $idx_{batch}, \tau, x \leftarrow enumerate(GT_{batch})$ **do**
5:    $dict_{idx}[idx_{batch}] = (\tau, \text{len}(dict_x[\tau]))$
6:    $dict_x[\tau]$.append($x$)           $\triangleright$ How to combine data of the same type
7:  **end for**
8:  **for** $\tau, X_\tau \leftarrow dict_x.items()$ **do**            $\triangleright \tau(key), X(values)$
9:    $\mathbf{x}_\tau = [\Psi[\tau](X_\tau), \text{onehot}(\tau).\text{repeat}(\text{len}(X_\tau))]$  $\triangleright$ Type Batch Convolution
10:    $dict_x[\tau] = \mathbf{x}_\tau$
11:  **end for**
12:  $output_{batch} = []$
13:  **for** $idx_{batch} \leftarrow 1...N$ **do**                $\triangleright Restoration$
14:    $\tau, idx_\tau \leftarrow dict_{idx}[idx_{batch}]$
15:    $\mathbf{x}_\tau = dict_x[\tau]$
16:    $output_{batch}$.append($\mathbf{x}_\tau[idx_\tau]$)
17:  **end for**
18:  **return** $Stack(output_{batch})$
19: **end function**
---

First, make two dictionary data structures. One is a dictionary($dict_x$) that stores data for each data type. The other is a dictionary($dict_{idx}$) that preserves the batch-index information of the data. We tour the mini-batch data in order and store the key is the type($\tau$) of data, and the value is the input data($x$). And the other dictionary stores the type and index.

Therefore, the input data are collected for each type, which is called $X_\tau$. And the input data become type-mini-batch data for each type and becomes batch-inputs for the feature extraction network of that type.

Let $\mathbf{x}_\tau$ denote the output of the feature extraction network. Finally, the $\mathbf{x}_\tau$ moves to the original batch index through the previously-stored batch index($dict_{idx}$).

## B  Compare Architecture: GTC & GTR

Table 3: Compare Architecture

| Name | GTC | GTR |
|---|---|---|
| Level layer | O | X |
| The number of **W** | The number of levels | 1 |
| Depth-limited | Fixed | Not fixed |
| The number of parameters | Use more | Use less than |
| Input size by level | Adjustable | Fixed |
| The special cases | FCNN, MLP | RNN(recurrent, recursive) |

In this section, we compare the network characteristics of GTC-series and GTR-series recursive models. In the case of GTC, there is a different $\mathbf{W}_{lv}$ for each level, expressing each level layer and the convolution of all information. On the other hand, in the case of GTR, the main difference is the recursive convolution with only one W as a cell. This is the most significant difference when comparing the networks.

Therefore, GTC can adjust the number of features by level; for example, the size of input features can differ between Levels 1 and 0. It is possible for GTC to train datasets by adjusting parameters at any level. On the other hand, GTR must be trained with the same input and hidden size.

Thus, we divided GTNN into two groups. Since GTC has a network in charge of each level, more parameters are needed, and it is appropriate for depth-limited GT. if the $F_{lv}$ of input size is 0 and the number of nodes is 1 in GTN, FCNN and MLP can be special cases of GTC as $\mathbf{IW}_{lv}\vec{x}^T$ and $(\mathbf{IW}_0..(\mathbf{IW}_{lv-1}(\mathbf{IW}_{lv}\vec{x}^T)))$. On the other hand, GTR traverses all GTNs with a GTR cell. Therefore it is possible to train even if the maximum depth is not fixed and fewer parameters are used.

in addition, If there is no sibling node in $\mathbf{GTN}_p$, RNN can be a special cases of GTR as $\mathbf{IW}[\vec{x}, \vec{h}]^T$. Consequently, the compare result in Table 3.

## C    USED MODELS

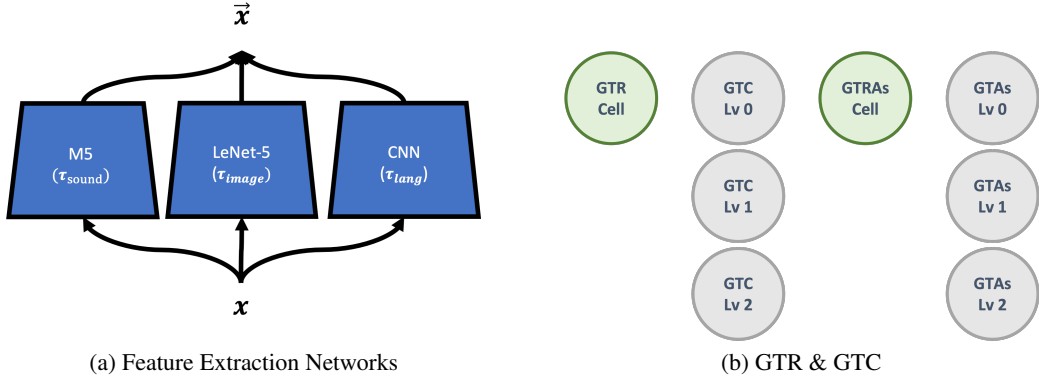

(a) Feature Extraction Networks          (b) GTR & GTC

Figure 6: Used Models

Table 4: Feature Extraction Networks for Image dataset

| | LeNet-5 | | | | | LeNet-5 for GTNN | | | | |
|---|---|---|---|---|---|---|---|---|---|---|
| Model | In | Out | kernel | stride | activation | In | Out | kernel | stride | activation |
| Conv2D | 1 | 6 | (5,5) | 1 | tanh | 1 | 6 | (5,5) | 1 | tanh |
| AvgPool2D | - | - | (2,2) | 1 | - | - | - | (2,2) | 1 | - |
| Conv2D | 6 | 16 | (5,5) | 1 | tanh | 6 | 16 | (5,5) | 1 | tanh |
| AvgPool2D | - | - | (2,2) | 1 | - | - | - | (2,2) | 1 | - |
| Conv2D | 16 | 120 | (5,5) | 1 | tanh | 16 | 120 | (5,5) | 1 | tanh |
| FC layer 1 | 120 | 84 | - | - | tanh | - | - | - | - | - |
| FC layer 2 | 84 | 10 | - | - | softmax | - | - | - | - | - |
| Zero padding | - | - | - | - | - | 120 | 128 | - | - | - |
| Final | - | 10 | - | - | - | - | 128 | - | - | - |

LeNet-5(LeCun et al., 1989) was used as the image feature extraction network. We create a dimension of 128 by applying zero padding to the extracted features without using the affine-layer(FC layers) and then forward it to GTNN.

Table 5: Feature Extraction Networks for Sound dataset

| | M5(Group Norm) | | | | | | M5(Group Norm) for GTNN | | | | | |
|---|---|---|---|---|---|---|---|---|---|---|---|---|
| Model | In | Out | kernel | stride | norm | activation | In | Out | kernel | stride | norm | activation |
| Conv1D | 1 | 128 | 80 | 4 | group 16 | relu | 1 | 128 | 80 | 4 | group 16 | relu |
| MaxPool1D | - | - | 4 | 1 | - | - | - | - | 4 | 1 | - | - |
| Conv1D | 128 | 128 | 3 | 1 | group 16 | relu | 128 | 128 | 3 | 1 | group 16 | relu |
| MaxPool1D | - | - | 4 | 1 | - | - | - | - | 4 | 1 | - | - |
| Conv1D | 128 | 256 | 3 | 1 | group 16 | relu | 128 | 256 | 3 | 1 | group 16 | relu |
| MaxPool1D | - | - | 4 | 1 | - | - | - | - | 4 | 1 | - | - |
| Conv1D | 256 | 512 | 3 | 1 | group 16 | relu | 256 | 512 | 3 | 1 | group 16 | relu |
| MaxPool1D | - | - | 4 | 1 | - | - | - | - | 4 | 1 | - | - |
| AdaptiveAvgPool1d | - | 1 | - | - | - | - | - | 1 | - | - | - | - |
| FC layer | 512 | 35 | - | - | - | log softmax | 512 | 128 | - | - | - | leaky relu |
| Final | - | 35 | - | - | - | - | - | 128 | - | - | - | - |

M5(Dai et al., 2017) was used as the sound feature extraction network. As described above(Sec.2.2), we used group normalization(Wu & He, 2018) without using Batch norm(Ioffe & Szegedy, 2015). We use the affine layer(FC layer) and deliver it to the GTNN in 128 dimensions. It could be implemented in torchaudio[1].

Table 6: Feature Extraction Networks for Natural language dataset

| Model | CNN | | | | | CNN for GTNN | | | | |
|---|---|---|---|---|---|---|---|---|---|---|
| | In | Out | kernel | stride | activation | In | Out | kernel | stride | activation |
| Conv2D | 1 | 100 | (3,3) | 1 | relu | 1 | 100 | (3,3) | 1 | relu |
| AvgPool2D | - | - | (2,2) | 1 | - | - | - | (2,2) | 1 | - |
| Conv2D | 1 | 100 | (4,4) | 1 | relu | 1 | 100 | (4,4) | 1 | relu |
| AvgPool2D | - | - | (2,2) | 1 | - | - | - | (2,2) | 1 | - |
| Conv2D | 1 | 100 | (5,5) | 1 | relu | 1 | 100 | (5,5) | 1 | relu |
| Concat | (300,400,500) | 1200 | - | - | - | (300,400,500) | 1200 | - | - | - |
| Dropout | - | - | - | - | - | - | - | - | - | - |
| FC layer | 1200 | 1 | - | - | - | 1200 | 128 | - | - | - |
| Final | - | 1 | - | - | - | - | 128 | - | - | - |

CNN(Kim, 2014) was used as the feature extraction network for natural language processing. We slightly modified the contents in this paper and combined them with GTNN. We did not use drop out because we wanted to emphasize the difference between transfer learning and general learning in the association model. The glove(Pennington et al., 2014) was used for the pre-trained word embedding network with 100 dimensions. We used an adam optimizer(Kingma & Ba, 2014), a learning rate of 0.001, a cosine annealing(T max=2, eta min=1e-05) for schedulers(Loshchilov & Hutter, 2016). the batch-size is 32 and the MNIST, Speech Command and IMDB classes are 10, 35, 2.

## D   HOW DO WE DESIGN THE STRUCTURE OF THE GRAPH TREE?

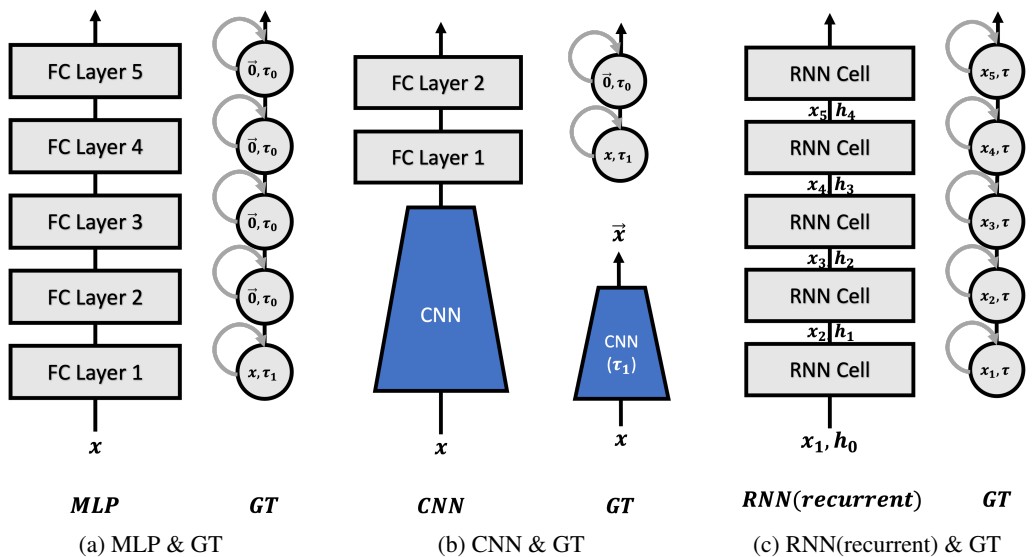

Figure 7: Model structure & Data structure 1

These architectures are the models underlying existing networks. First, Fig.7(a) means a multi-layer perceptron model. If we express this architecture as GT and learn with GTC, which has input only in leaf node, it becomes the same operation process.

---

[1] https://pytorch.org/tutorials/intermediate/speech_command_recognition_with_torchaudio_tutorial.html

Second, Fig.2(b) means a CNN model. Using the CNN network for the feature extraction network part with GTC will be the same operation.

Third, Fig.7(c) means a recurrent neural network model(RNN). Using a tree with only one child to express sequence and learn with GTR will be the same operation.

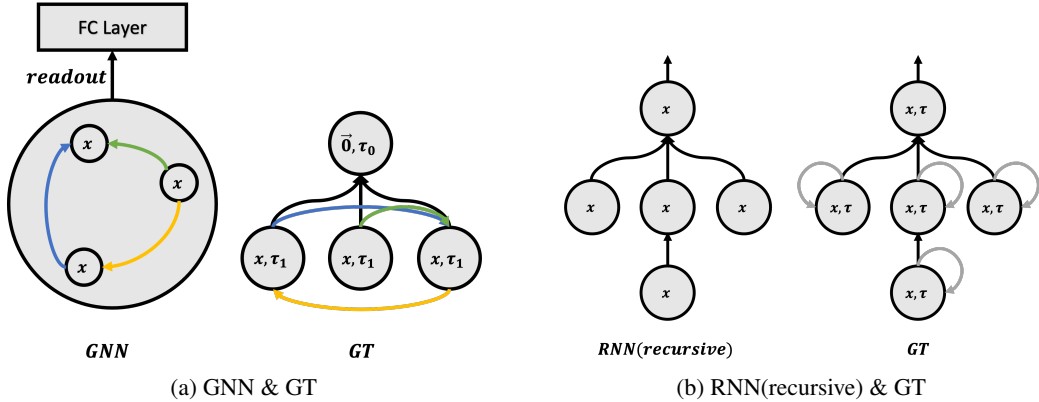

(a) GNN & GT          (b) RNN(recursive) & GT

Figure 8: Model structure & Data structure 2

These models are architectures that learn relational and hierarchical information. First, Fig.8(a) means a GNN model. The same delivery process is obtained if the relationship and inputs are expressed as relationship and inputs among sibling nodes of the graph tree with GTC.

Second, Fig.8(b) means a recursive neural network model(RNN). If we learn using an identity matrix with GTR, it will be the same delivery process.

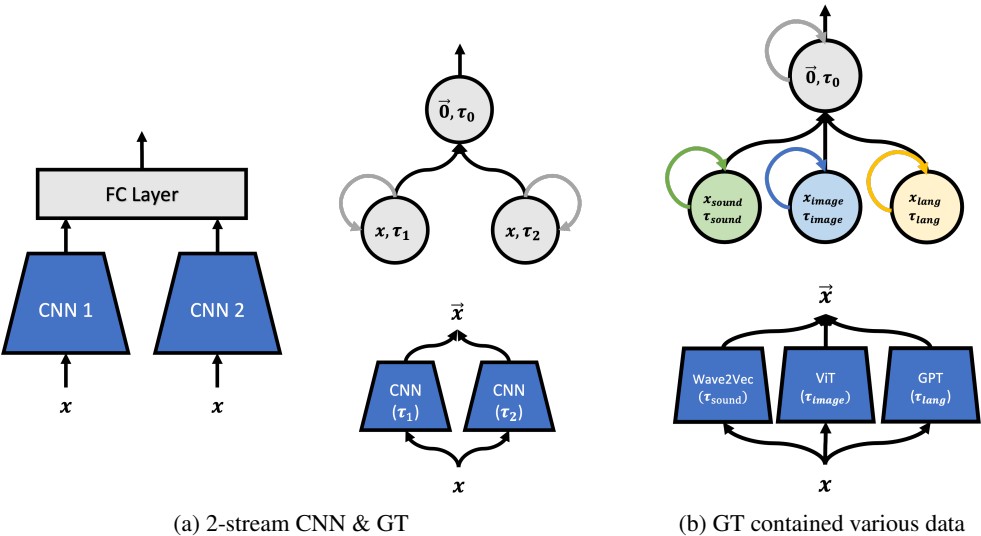

(a) 2-stream CNN & GT          (b) GT contained various data

Figure 9: Model structure & Data structure 3

These networks are structures in which each feature extraction process is combined. If we design feature extraction networks for each type and construct a graph tree with two or three children Fig.9(a,b), it will be the same delivery process.

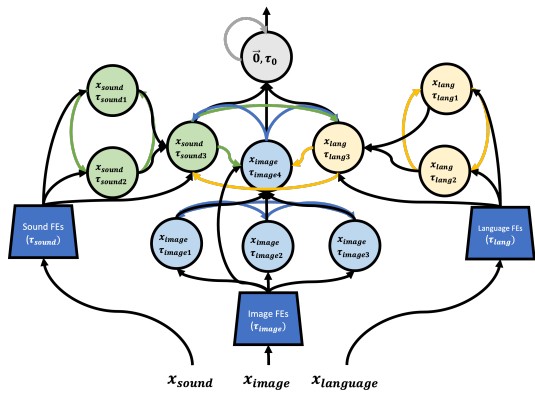

Figure 10: Ideal GT

Finally, this structure is an ideal graph tree structure (Fig.10). The feature extraction network corresponds to the sensory organ. And the feature extraction process($\Psi$) is selected according to the type($\tau$) of data x($x$), and x becomes vector $\vec{x}$.

And information integration occurs. We call it an association network. This network serves to embed various extracted information into one vector($\vec{h}_{root}$) by applying relationship and hierarchical information.

At this time, expressing a tree structure that learns well for each type of data or task is the same as designing the architecture of the network.

Therefore, instead of using fixed layers to learn according to the flow of information, build a graph tree to learn according to the flow of information. It will enable the network to learn networks that can handle multiple tasks or multiple datasets simultaneously, not just specific tasks or specific datasets. Because various architectures can be expressed by data.

Visual information is input when we open our eyes, and information is not input when we close our eyes. It can be expressed when there is or does not have image information in the graph tree.

In addition, it could express a structure such as V1, V2, V3, V4, V5/MT in visual cortex, and a graph tree has hierarchical information and relational information. The grammar of natural languages can also be expressed as a tree parser.

Voice information can also be extracted from MFCC algorithms etc. And all information is integrated and embedded in association networks of Graph Tree Neural Networks.

## E    THE EXPERIMENT 3 : INDIVIDUAL & SIMULTANEOUS

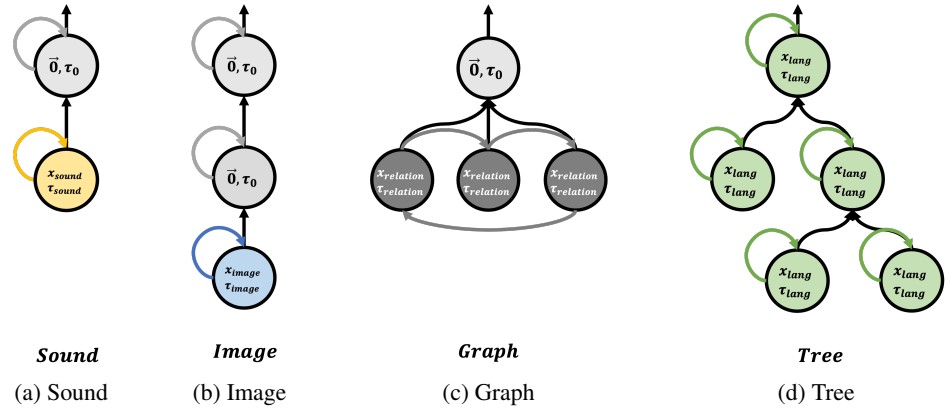

Figure 11: Graph tree datasets of the experiment 3

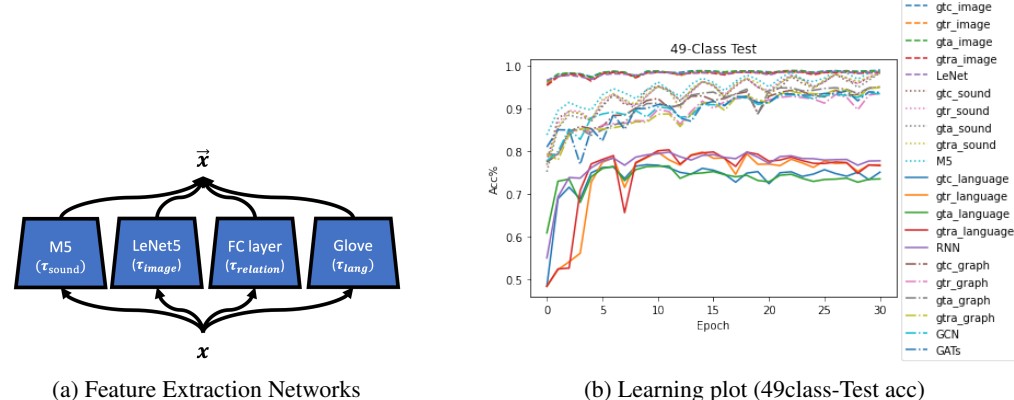

    (a) Feature Extraction Networks              (b) Learning plot (49class-Test acc)

Figure 12: Feature Extraction Networks & Learning plot

As a third experiment, we learned datasets of various structures with GTNN. Data in the form of images, sound data, language data(Socher et al., 2013) in the tree structure, and data in the graph structure(Dou et al., 2021) were learned at the same time, and the results are as follows. All of the performances at this time were similar to those of existing networks.

In addition, in the case of GTR and GTRAs, it can be seen that the learning speed for language data is lower than Recursive Neural Networks.

The reason is that the size of language dataset is relatively less than others. Therefore, since the language model has a relatively less influence on loss, it seems to have been pushed back in the optimization process.

In the case of GTC and GTAs, the result of overfitting in the language model is shown. This is a natural result because the layers of tree data are learned differently for each level.

As a result, we proved that learning to express various information delivery structures with one network cell does not significantly affect performance. When we stop learning this network at 30

Table 7: the result of Experiment 3

|  | 49(=10+35+2+2) class | | | |
| --- | --- | --- | --- | --- |
| Model | MNIST | SC | SST | UPFD-GOS |
| LeNet-5 | 98.52 | - | - | - |
| M5 (Group Norm) | - | **98.37** | - | - |
| RNN(Recursive) | - | - | **77.69** | - |
| GCN | - | - | - | 93.67 |
| GATs | - | - | - | 93.93 |
| GTC | **98.88** | 97.18 | 73.30 | 94.74 |
| GTAs | 98.80 | 96.67 | 73.39 | **94.85** |
| GTR | 98.68 | 97.46 | 76.74 | 93.15 |
| GTRAs | 98.35 | 97.44 | 76.79 | 94.67 |
| class count | 10 | 35 | 2 | 2 |
| type | image | sound | tree | graph |

epochs, the performance in test dataset is in the table7.

