# OpenReview forum: "Graph Tree Neural Networks"
_ICLR.cc/2022/Conference — ICLR 2022 Submitted_

### Official Review · Reviewer_sfjx · 2021-11-02

**Correctness:** 2
**Technical Novelty And Significance:** 2
**Empirical Novelty And Significance:** Not applicable
**Recommendation:** 1
**Confidence:** 3

**Main Review:**

## What I like about this paper

- The idea of unifying different architectures (RNN, Convolution, MLP) into one big neural network organized as a graph tree.

- Dealing with multiple types of input modality is an important research direction.


## Concerns

- The paper is hard to follow. Several sentences are not well-formatted. I found it hard to extract the relevant contributions and claims from this paper.

- I'm not convinced by the empirical results presented in section 3. All the numbers look very close to each other and might be within the error margin. I would have liked to have the source code, to test it myself.

- On page 7, the authors mention the image GT has three nodes since "the occipital lobe is the farthest from the frontal lobe in the human brain". I find that justification dubious as longer axonal connections don't mean more processing. Citations would be needed to back up this claim.

- A related work section is missing? I doubt there is no other work that can be related to this one. What are the limitations of let's say Graph Neural Network compared to Graph Tree Neural Network?

- P.1: what does it mean to "freely integrated"?
- P.1: there's a claim about "existing networks are designed ... layers are fixed" which is not true. Several papers describe approaches to adapt how information is processed in dynamic ways, e.g. ResNet [1], Adaptive Computation Time [2], etc.

#### References
- [1] He, Kaiming et al. “Deep Residual Learning for Image Recognition.” 2016 IEEE Conference on Computer Vision and Pattern Recognition (CVPR) (2016)
- [2] Graves, Alex. “Adaptive Computation Time for Recurrent Neural Networks.” ArXiv abs/1603.08983 (2016)

### Minor

- This paper has heavy usage of acronyms. For the sake of clarity in the paper, I'd use the full names rather than the acronyms when talking about GT, GTN, GTNN, GTR, and GTC. Otherwise, I'd try to reduce their usage as much as possible.

- I'm concerned by the name used to refer to the proposed approach. To me, Graph Tree Neural Networks means a neural network capable of taking graphs shaped as trees as input.

-----
### Typos
- A lot of sentences don't start with an uppercase letter.
- Missing whitespaces before several opening parentheses.


**Summary Of The Paper:**

In this paper, the authors propose Graph Tree Neural Network (GTNN) a new learning model that is structured as a graph tree (i.e., a tree with links between siblings) where each node has to process the data coming from its children. GTNN can take as input data of various types where each node will be processing the data differently according to the data type. The authors describe how (de)convolution and recurrent neural networks can be achieved within GTNN. Experiments were conducted on some classic deep learning datasets of three different data types: images (MNIST), language (IMDB), and sound (Speech Command). The goals of those experiments were two see whether GTNN can process various datasets of different types and if GTNN can produce an output vector that retains all the information coming from the different data types.

**Summary Of The Review:**

I believe designing novel architectures inspired by the human brain is an interesting and important research direction. However, in its current state, this paper was not able to convince me that the GTNN architecture was one worth pursuing. To be honest, it wasn't clear to me what were the actual contributions of the paper, and how the experiment results are backing up the claims made in this paper. For those reasons, I recommend rejecting this paper for ICLR.

---

> ### Author Response · Authors · 2021-11-11
> **Thank you for your review!**
>
> Thank you for taking the time to review!
>
> 1. I added figures to the appendix, but I'm trying to revise the article.
>
> 2. I didn't refine the code, so I was tried to develop and distribute a library later.
>
> 3. In this study, we compared and introduced the brain to intensively explain the brain's association area, the motivation of GTNN.
>
> To help understand, we similarly used the image of the brain and the structure of the tree.
> But the phrase seems to be misleading. Therefore, I revised the sentence.
>
> 4. Related works section has been combined with the introduction section due to page restrictions.
> The difference between GNN and GTNN is that feature extraction networks are added, and the convolution path is simplified by DFC and DFD transforming into a tree structure.
> Additionally, it can learn by expressing the relationship between sibling nodes in all level processes with a graph tree, not a single graph or tree.
>
> In summary, We have alleviated the problem of fixed delivery flow based on the tree. And it simplified routes by Depth-First-Convolution and Depth-First-Deconvolution.
>
> P.1 The content was added as two images in the supplementary material (freely_integrated_means_1,2.png).
>
> ResNet is a model specialized in the Feature Extraction process because it is performed based on VGG-Net and CNN.
> However, this model(GTNN) is closer to an association model than feature extraction.
>
> The sentence means that models such as 2-stream and 3-stream CNNs go through FCNNs through a concatenation process.
> I am saying that this structure has a fixed layer.
> +And I will change the name of this paper.

---

### Official Review · Reviewer_Svha · 2021-11-02

**Correctness:** 2
**Technical Novelty And Significance:** 2
**Empirical Novelty And Significance:** 2
**Recommendation:** 1
**Confidence:** 3

**Main Review:**

From what I understand, I really like the research idea and direction of the paper. Unfortunately, I had extreme difficulties understanding it because the paper lacks a good structure and a fluent narrative (details below). Therefore, I cannot properly comment on the paper's strengths and weaknesses. Instead, I list the questions I had while reading the paper.

* Apart from having a new architecture, I do not understand the exact problem setting of the paper: is it supposed to be multi-modality? Multi-task learning? An analysis of the relationship of brain-like learning and artificial learning? This is important, because a proper problem setting would determine the choice of baseline architectures and experiments. As things are, the paper feels "ungrounded".
* In the paper's claims, several related works are not mentioned. In particular regarding dynamic computation graphs and dynamic resource allocation, the work around conditional computation (mixture of experts; more relevant, the "Routing" literature (Routing Networks and the Challenges of Modular and Compositional Computation) come to mind)
* Terminology throughout the paper is strange; instead of "inputs" and "outputs" or "predictions" we have "start" and "endpoints". I understood this eventually, but it created some confusion for me in the beginning.
* Somewhat related to the first point, I'm missing a "big picture" overview of what the proposed architecture tries to achieve, and why the proposed architecture is good at achieving this. This is particularly relevant for section 2: I wish for an overview at the beginning of each subsection (2.1-2.5), describing what the respective component is mean to achieve, how it does so, and why it was designed in this particular way.
* Unfortunately, my confusion continues in the experimental section. I suspect that the experiments are non-standard, but I cannot actually tell, because the experimental setup is not described anywhere in detail.
* I am not perfectly sure on the experiments. However, I still want to point out that the baselines are out of date. That may be fine, but then the choice of these baselines needs to be motivated.

**Summary Of The Paper:**

The paper introduces a new neural network architecture called a "Graph Tree Neural Network". This architecture is inspired by several properties of the human brain: 1. it's multi-modal; 2. it allows for cross-input reasoning; 3. it can adapt its internal structure to the current input; 4. it can adapt the amount of computation per input; and 5. the human brain is larger than most neural networks.

The architecture itself relies on mode-specific representation components: later, in the experiments, they introduce targeted architectures for images, speech and text. These representations are then consumes by a convolutional logic that allows full communication within the different components of the network. Finally, these components can be recursively combined to yield a full graph tree neural network.

This architecture is then evaluated on several domains covering speech, vision and nlp datasets, showing consistent improvement over the selected baselines.

**Summary Of The Review:**

Overall, I like the goal and the ideas behind the paper. Unfortunately, the paper is extremely unclear in its presentation: I could not put individual sections into context, I do not know what the goal of the paper is, and I don't understand the results. Therefore, I cannot recommend this paper for acceptance.

---

> ### Author Response · Authors · 2021-11-11
> **Thank you for your review!**
>
> Thank you for taking the time to review!
>
> 1. In this paper, multi-domain and brain-like learning are the topics.
> And the future works introduce multi-modal, task learning.
> We present the problems found by comparing the brain and networks as the beginning of the series in this paper.
>
> 2. I'll add it in the next version.
>
> 3. The reason is that I wanted to talk about the convolution path of DFC and DFD.
> And I thought input and output were terms from a data perspective and starting point and ending point were terms related to the path.
>
> 4. Section 2 will be revised in the next version.
>
> 5. The used model information is described in Appendix C.
>
> 6. The reason for using this model is that neural network models in the image field recently used batch normalization.
>
> But, the mini-batch size varies every time during the feature extraction process for each type, we recommend group normalization or weight standardization that is less affected by the batch size. and I described this in sec 2.2. and Appendix A.
>
> For example, if the GT mini-batch size is 32, image GT: 15, sound GT: 10, and language GT: 7, and the size changes every time(15(type a)+10(type b)+7(type c) = 32(mini-batch)).
>
> So, in order not to change the existing model as much as possible, we used LeNet-5, which does not use batch normalization.

---

### Official Review · Reviewer_oPzs · 2021-11-03

**Correctness:** 2
**Technical Novelty And Significance:** 2
**Empirical Novelty And Significance:** Not applicable
**Recommendation:** 1
**Confidence:** 4

**Details Of Ethics Concerns:**

N.A.

**Main Review:**

Generally, I don't (and shouldn’t) reject a paper because of writing.
However, the writing of this paper is too hard to follow.
Among hundreds of papers I have reviewed, it is the most distracting one.
Even if there are significant technical contributions, I still cannot accept this paper because of the writing.
However, the idea of applying a tree-like structure to generalize neural networks is still interesting.
I am sorry that I cannot complete the reading because of the writing.

**Summary Of The Paper:**

It **seems** that this paper tries to propose a general neural network architecture that is composed of multiple layer types based on a tree structure.

**Summary Of The Review:**

Writing is too hard to follow.

---

> ### Author Response · Authors · 2021-11-11
> **Thank you for your review!**
>
> Thank you for taking the time to review!
>
> If possible, may I know which section was not easy to read?
> I want to complete this study, and I will revise it by referring to the comments.

---

### Author Response · Authors · 2021-11-11
**to reviewers**

Thank you for the reviews!

First, this paper had a lot to explain.
(motivation) human brain, (data structure) graph tree, (propagation algorithm) Depth-first Convolution, Depth-first Deconvolution, (Networks) graph tree recursive, convolution and attention versions.

In particular, the DFC and DFD processes are fundamental algorithms in this study and are a propagation methodology that simultaneously learns relational, hierarchical, and feature extraction 'networks'.

Therefore, we expressed what I think as pictures in the appendix to be easily understood.
If it was not easy to understand, please check the 'appendix figures'.

And I tried to add an experimental section, but the 9-page limit was painful for me. So I removed some explanations, and I think that caused the problem.

------------ (The subject of this study) ------------

Existing network models use a fixed layer so that they can express only one structure.

However, this study does not express layers but express and learn various information delivery structures in a tree structure.

Typically, N-stream models (ex: 2-stream CNN) => two CNNs extract features from an image and concatenate the outputs.

On the other hand, this study has two child nodes that express information in a tree structure that delivers it to the parent node.

Therefore, we separated existing models into two structures, feature 'extraction' networks and 'association' networks (GTNNs).

Expressing the path in which features are combined and propagated in trees will be multi-domain and multi-modal deep learning.

—————— (The experiment of this study) ——————

The three experiments were organized as follows.
1. combining feature extraction networks + association networks (multi-domain)
2. the output(root vector) is whether It contained all information (It shows scalability to multi-modal)?
3. could datasets with various structures(image, sound, tree, graph) be learned with GTNN(multi-domain)?

Experiment 3 has just been added.

So if possible, I would like to get the reviews this time and complete it well next time.

Thank you!

---

### Decision · Program_Chairs · 2022-01-20

**Decision:**

Reject

**Comment:**

First, I would like to thank all the reviewers for their efforts in reading and understanding this paper. I tried to read the paper as well and I also find it's really difficult (if possible) for me to understand the ideas presented here. The most important task in writing a paper (as Reviewer Svha also suggested in his/her review) in the field of machine learning is to explain to your peers what is the problem you are trying to solve and how you solve (or partially solve) that problem. I think there is a consensus among the reviewers that the paper did not do a great job of that. I am not questioning the quality of the idea or the research here, but I think the paper here will need to do a significantly better job here in explaining the idea before it can be a good ICLR publication.